civil engineering/energy/engineering geology

nuclear energy, high-level radioactive waste, granite, fracture toughness, temperature duration

**Authors for correspondence:**
Gan Feng
e-mail: fenggan@whu.edu.cn
Yong Kang
e-mail: kangyong@whu.edu.cn

# Fracture failure of granite after varied durations of thermal treatment: an experimental study

Gan Feng[1,2,4], Yong Kang[1,3] and Xiao-chuan Wang[1,3]

[1]Hubei Key Laboratory of Waterjet Theory and New Technology, [2]The Key Laboratory of Safety for Geotechnical and Structural Engineering of Hubei Province, School of Civil Engineering, and [3]School of Power and Mechanical Engineering, Wuhan University, Wuhan 430072, People's Republic of China
[4]Department of Energy and Mineral Engineering, Pennsylvania State University, University Park, PA 16802, USA

GF, 0000-0002-7110-4427; YK, 0000-0003-0991-5314

Energy extraction from nuclear materials produces high-level radioactive waste. In geological nuclear waste storage repositories, the decay of radioactive elements generates heat, exposing the reservoir rocks to high-temperature conditions for long periods. To explore the effects of these conditions, this study examines the ability of granite to resist fracturing after thermal treatment for 10 h, 10 days, 30 days and 60 days. The results show that the fracture toughness of the granite remained basically unchanged for up to 10 days of thermal treatment. After thermal treatment for 60 days, the mode I, mode II and mixed-mode (I + II) fracture toughness decreased by 15.39%, 18.09% and 15.17%, respectively, compared with samples heated for 10 h. The change trends of the ability of granite to resist tensile, shear and mixed (tensile + shear) failure with an increased thermal treatment duration were basically consistent. Moreover, there was little change in its brittle fracturing characteristics with an increase in heating duration. Changes caused to the internal microstructure of the granite by high temperature were ongoing even up to 60 days.

## 1. Introduction

With the ongoing depletion of traditional energy sources, the development of nuclear energy has been receiving growing attention. However, nuclear energy production results in high levels of nuclear waste, and the safe disposal of this waste is of major importance given the threat it poses to the health and living environment of human beings. At present, geological

**Figure 1.** Geological disposal of high-level radioactive waste (HLW) [1].

storage, as shown in figure 1, is one of the most effective methods of disposing of high-level nuclear waste [1]. The rock surrounding such repositories is generally granite. Over time, the decay of radioactive elements gradually produces heat that radiates and causes the temperature of the surrounding rocks to rise. There are predictions that the released heat can cause the temperature of the granite to rise to 200–300°C [2,3] and even reach 500°C in some areas. If the effects of long-term exposure to such high temperatures should cause the rock to fail, the nuclear waste will leak, which will cause major pollution in the underground environment. There is substantial evidence that the mechanical properties of granite change with exposure to high temperatures. Kumari *et al.* [4] found that, as temperature increases, the strength and shear parameters of Australian Strathbogie granite first increase and then decrease. They also noted that the Mohr–Coulomb criterion is not suitable for modelling rock under high temperature and high pressure. Real-time changes to the permeability of granite under high temperature were studied by Feng *et al.* [5]. The results showed 300°C to be the threshold temperature ($T_C$) of granite permeability change and that this change results from coupled thermal and mechanical effects. Nasseri *et al.* [6] studied the fracture toughness, elastic wave velocities and microcrack density of Westerly granite, and found that thermal cracking not only caused a decrease in its mechanical strength but also affected its dynamic elastic properties. In particular, the normalized compressional P-wave velocities showed a very similar decrease trend to that of fracture toughness. Lin [7] further showed that permanent strain, reflecting the cumulative amount of newly generated microcracks and the opening of pre-existing microcracks, appeared distinctly at a temperature between 100°C and 125°C in Inada granite. Wang *et al.* [8] studied the physical properties and brittle strength of heat-treated granite, finding that its permeability was four to five orders of magnitude larger than that of an intact sample. Zhao *et al.* [9] further showed that the permeability of granite increases significantly when heated above a critical temperature.

Under the action of high temperature, mineral particles expand and produce thermal cracks. Cracks in complex geological environments may cause instability and may penetrate to form macroscopic cracks. The problems thus raised are studied with the aid of fracture mechanics, which is commonly applied in the field of underground engineering. Fracture toughness is an important indicator of the ability of rock to resist fracture failure and is therefore useful for assessing the stability of geological nuclear waste storage sealing. The fracture characteristics are an important index to study the strength and deformation behaviour of rock mass in rock engineering [10]. When a crack is subjected to complex loading, it can fracture in three basic modes, namely mode I, mode II and mode III, as shown in figure 2. Any combination of these basic fracture modes becomes a mixed mode.

In geological nuclear waste storage engineering, a crack is first formed in the place where the stress is most concentrated. These cracks are usually subject to tensile stress, shear stress and mixed-mode stress; thus, mode I, mode II and mixed-mode (I + II) fractures occur. The study of the fracture toughness of granite after exposure to high temperature is beneficial for evaluating the stability of underground storage of nuclear waste. Geological storage of nuclear waste in granite is a long-term process. Many studies currently use a 1 h, 2 h or 8 h temperature exposure duration [11–13]. Studies employing longer term temperature exposure are scarce. In geological nuclear waste storage, granite is in a long-term, continuous high-temperature environment, and it is necessary to study the changes that this

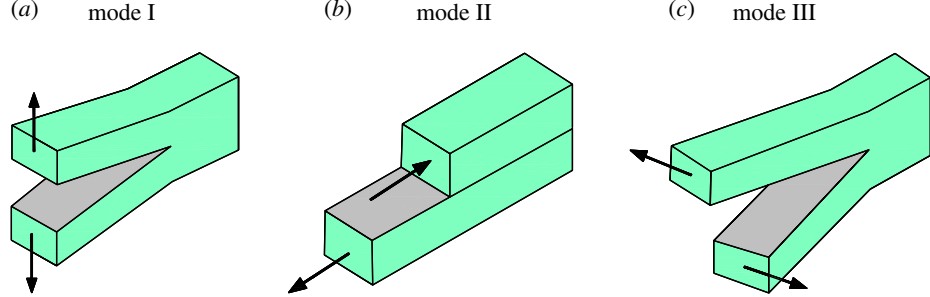

**Figure 2.** ($a$–$c$) Three basic fracture modes.

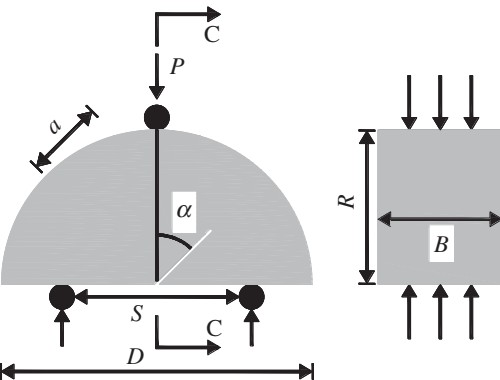

**Figure 3.** Geometry of an SCB specimen and schematic of the loading arrangement.

causes to its physical and mechanical properties. Therefore, in this study, granite was heated to a high temperature of 300°C and was kept at that temperature for 10 h, 10 days, 30 days or 60 days. The pure mode I, mode II and mixed-mode (I + II) fracture toughness of the granite was then tested, and the effect of the duration of temperature exposure on the ability of the rock to resist fracture failure was studied.

## 2. Fracture toughness determination

The International Society for Rock Mechanics (ISRM) recommends several methods for testing the fracture toughness of rock [14–16]. This paper uses the semicircular bend (SCB) method recommended by the ISRM in 2014, which is shown in figure 3.

In the figure, $R$ is the sample radius, $B$ is the thickness of the SCB sample, $a$ is the length of the pre-notch, $S$ is the span between supports, $P$ is the load at the evaluation point, $\alpha$ is the angle of the pre-notch and $D$ is the diameter of the SCB sample. Different combinations of the above parameters can be used to estimate the mode I, mode II and mixed-mode (I + II) fracture toughness [17]. This study sets $S/D = 0.61$ and $a/R = 0.5$. With $a/R$ and $S/D$ fixed, any loading mode can be achieved by changing the angle of the pre-notch to the loading direction. When the angle of the pre-notch is $\alpha = 0°$, fracturing will be mode I. When the angle of the pre-notch is $\alpha = 54°$, it will be mode II. Other angles will lead to mixed-mode (I + II) fracture. In this study, $\alpha = 30°$ is selected to explore mixed-mode (I + II) fracture. A schematic diagram showing the SCB designs for loading to produce mode I, mode II and mixed-mode (I + II) fracture is presented in figure 4.

When the rock is subjected to an external load, stress concentration occurs first at the crack tip, which becomes unstable when the concentrated stress exceeds its strength. The crack will expand through the area made unstable by the stress concentration to the area where the external load is applied. In terms of fracture mechanics, the crack tip is unstable because the stress intensity factor has reached a critical value. The critical stress intensity factor can be calculated with the following formulae [18]:

$$K_{\mathrm{I}} = \frac{P}{DB}\sqrt{\pi a}\, Y_{\mathrm{I}} \tag{2.1}$$

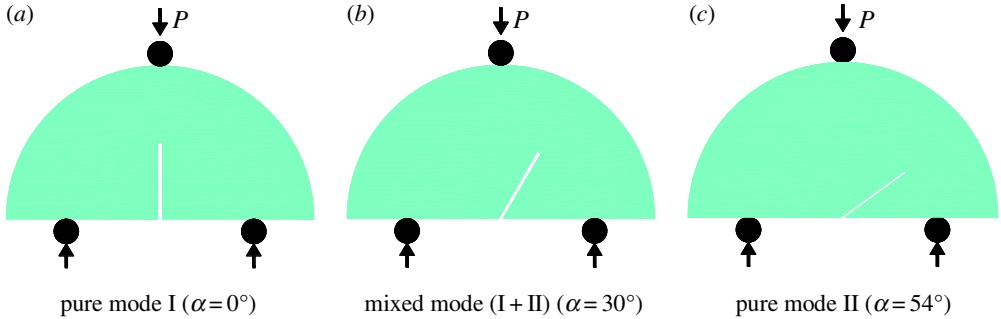

pure mode I ($\alpha = 0°$)       mixed mode (I + II) ($\alpha = 30°$)       pure mode II ($\alpha = 54°$)

**Figure 4.** ($a$–$c$) Angle designs to test fracturing by different modes.

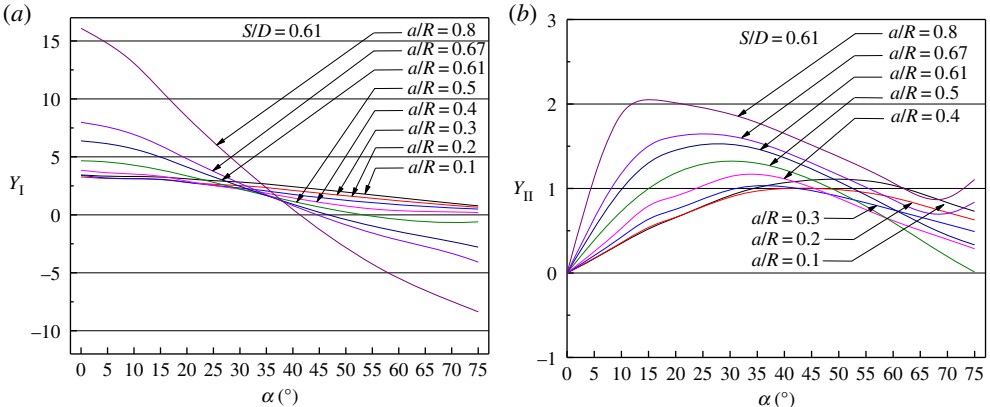

**Figure 5.** Normalized stress intensity factors $Y_I$ and $Y_{II}$. ($a$) Mode I component and ($b$) mode II component (curve data from Lim et al. [18] and Ayatollahi & Aliha [22]).

and

$$K_{II} = \frac{P}{DB}\sqrt{\pi a}\, Y_{II}. \tag{2.2}$$

Usually, the fracture toughness of rock can be characterized by the effective stress intensity factor ($K_{eff}$), as follows [19–21]:

$$K_{eff} = \sqrt{K_I^2 + K_{II}^2}. \tag{2.3}$$

In equations (2.1)–(2.3), $K_I$ is the mode I stress intensity factor and $K_{II}$ is the mode II stress intensity factor. $Y_I$ is the normalized stress intensity factor for mode I and $Y_{II}$ is the normalized stress intensity factor for mode II; these are functions of the ratio between the pre-notch length and sample radius $a/R$, the ratio between the span between the supports and the sample diameter $S/D$ and the pre-notch inclination angle $\alpha$. The values of $Y_I$ and $Y_{II}$ in the SCB test were studied by numerical simulation [18,21,22], giving the results shown in figure 5.

# 3. Sample preparation and fracture toughness tests

## 3.1. Sample preparation

The granite used in this experiment came from Suizhou, Hubei Province, China. Its mineral composition and content were assessed using X-ray fluorescence spectroscopy, X-ray diffraction (XRD) and petrographic thin sections. The XRD pattern obtained is shown in figure 6. The main minerals in the Suizhou granite are plagioclase (50–65%), quartz (25–35%) and potash feldspar (5–10%). There is a small amount of biotite and some other minerals that are difficult to distinguish. The main physical and mechanical properties of granite are determined by three minerals: plagioclase, quartz and potash feldspar.

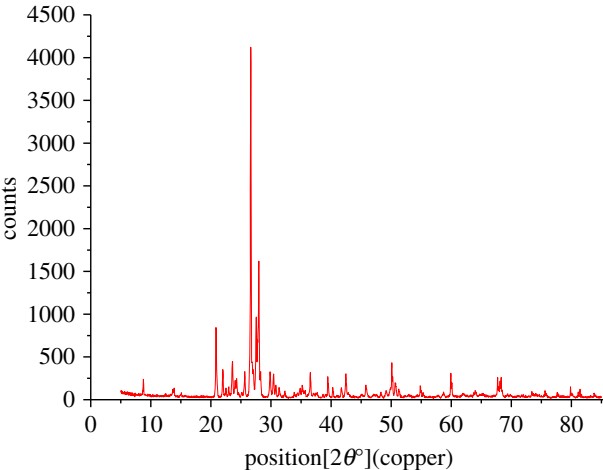

**Figure 6.** XRD spectrum.

(*a*)    cylindrical samples

(*b*)    disc samples

(*c*)    heating in a muffle

(*d*)    SCB sample

**Figure 7.** SCB sample processing.

A large block of granite was collected at the site and was drilled to extract cylindrical samples with a diameter of 50 mm and a height of 100 mm. The cylindrical samples were then cut into 20-mm-thick discs. Each disc was machined into a standard semicircular disc sample.

The temperature was set at 300°C for the experiment. The semicircular disc samples were placed in a smart muffle furnace and heated at a rate of 5°C min$^{-1}$. After the set temperature was reached, the temperature was maintained for the set heating duration, which was 10 h, 10 days, 30 days or

(a)    servohydraulic test system        (b)    loading configuration

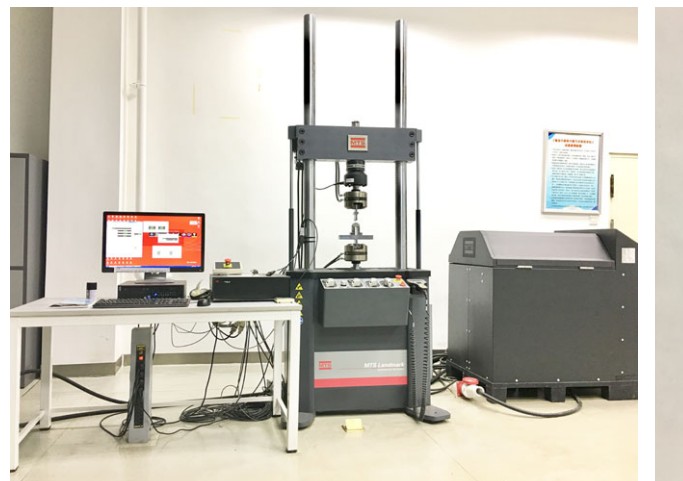 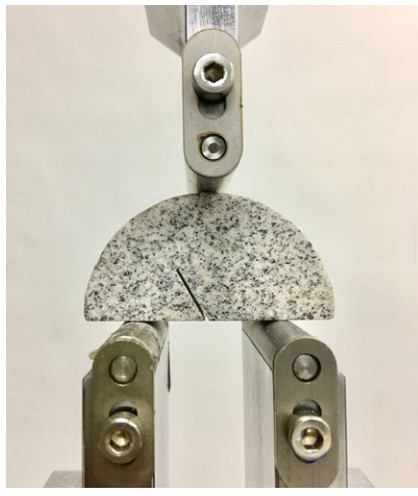

**Figure 8.** Experimental set-up for SCB tests.

60 days. After the thermal treatment had been completed, the heating was stopped, and the samples were allowed to cool naturally to room temperature.

After the thermal treatment was completed, pre-notches of a certain length were cut into all of the specimens with a precision bench drill. The pre-notch was then ground with a steel wire saw to make it straight, reducing the curvature of the crack tip and resulting in a straight, 12.5-mm-long pre-notch. The angle between the pre-notch and the vertical direction was $0°$, $30°$ and $54°$, respectively, to obtain SCB specimens suitable for the different loading modes. The standard SCB sample preparation process is shown in figure 7.

## 3.2. Fracture toughness tests

A three-point bending test was performed on the SCB samples, and the resulting data were automatically recorded by the computer. The experiment was carried out on the MTS experimental machine at the School of Civil Engineering at Wuhan University, China, using the displacement loading mode, with which the loading rate is $0.002\ \mathrm{mm\ s^{-1}}$. Figure 8 shows a fracture toughness test in progress. Three or four identical samples were set under each combination of variables as controls. Each sample was numbered in the form 'A−B−C' where, for example, '10d−0−3' indicates the third sample with a pre-notch angle of $0°$ that had received a 10 day thermal treatment.

# 4. Experimental results

## 4.1. Fracture toughness

Since sample processing always introduces errors, the actual size of each sample was measured and calculations were made according to the actual size. All the data obtained were brought into equations (2.1)−(2.3), and the effective stress intensity factor of the granite was calculated. The results are listed in table 1.

Figure 9 shows plots of the calculated mode I, mode II and mixed-mode (I + II) fracture toughness of the granite versus the duration of thermal treatment.

It can be seen from figure 9a that the mode I fracture toughness of the granite is $1.0111\ \mathrm{MPa\cdot m^{1/2}}$ after thermal treatment for 10 h. When the temperature duration is 10 days, the fracture toughness of the granite is $1.0386\ \mathrm{MPa\cdot m^{1/2}}$, a very small increase of 2.72% compared with the 10 h temperature duration. After heating for 30 days, the mode I fracture toughness is $0.9571\ \mathrm{MPa\cdot m^{1/2}}$, 7.85% lower than when the temperature was maintained for 10 days. With 60 days of heating, this value dropped to $0.8554\ \mathrm{MPa\cdot m^{1/2}}$, a decrease of 10.62% compared with 30 days. Therefore, the mode I fracture toughness of the granite was negligibly affected up to 10 days of thermal treatment, but it gradually reduced when heated for longer periods.

Figure 9b shows that the pure mode II fracture toughness of granite was $0.3901\ \mathrm{MPa\cdot m^{1/2}}$ at both 10 h and 10 days of thermal treatment. When the heating duration exceeded 10 days, there was a

**Table 1.** Effective stress intensity factor of granite (MPa · m$^{1/2}$).

| specimen | loading type | $K_I$ | $K_{II}$ | $K_{eff}$ |
|---|---|---|---|---|
| 10h–0–1 | mode I | 0.8909 | — | 0.8909 |
| 10h–0–2 | mode I | 0.9374 | — | 0.9374 |
| 10h–0–3 | mode I | 1.0361 | — | 1.0361 |
| 10h–0–4 | mode I | 1.1799 | — | 1.1799 |
| 10h–30–1 | mixed mode | 0.5467 | 0.3235 | 0.6352 |
| 10h–30–2 | mixed mode | 0.7713 | 0.4564 | 0.8962 |
| 10h–30–3 | mixed mode | 0.7253 | 0.4292 | 0.8428 |
| 10h–54–1 | mode II | — | 0.4421 | 0.4421 |
| 10h–54–2 | mode II | — | 0.3819 | 0.3819 |
| 10h–54–3 | mode II | — | 0.3464 | 0.3464 |
| 10d–0–1 | mode I | 0.9396 | — | 0.9396 |
| 10d–0–2 | mode I | 1.0697 | — | 1.0697 |
| 10d–0–3 | mode I | 1.1065 | — | 1.1065 |
| 10d–30–1 | mixed mode | 0.6277 | 0.3715 | 0.7294 |
| 10d–30–2 | mixed mode | 0.7260 | 0.4296 | 0.8436 |
| 10d–30–3 | mixed mode | 0.7423 | 0.4393 | 0.8626 |
| 10d–54–1 | mode II | — | 0.3404 | 0.3404 |
| 10d–54–2 | mode II | — | 0.4101 | 0.4101 |
| 10d–54–3 | mode II | — | 0.4307 | 0.4307 |
| 30d–0–1 | mode I | 0.9012 | — | 0.9012 |
| 30d–0–2 | mode I | 0.9277 | — | 0.9277 |
| 30d–0–3 | mode I | 1.0423 | — | 1.0423 |
| 30d–30–1 | mixed mode | 0.5918 | 0.3502 | 0.6877 |
| 30d–30–2 | mixed mode | 0.6207 | 0.3673 | 0.7213 |
| 30d–30–3 | mixed mode | 0.6876 | 0.4069 | 0.7990 |
| 30d–54–1 | mode II | — | 0.3136 | 0.3136 |
| 30d–54–2 | mode II | — | 0.3295 | 0.3295 |
| 30d–54–3 | mode II | — | 0.4127 | 0.4127 |
| 60d–0–1 | mode I | 0.7983 | — | 0.7983 |
| 60d–0–2 | mode I | 0.8460 | — | 0.8460 |
| 60d–0–3 | mode I | 0.9220 | — | 0.9220 |
| 60d–30–1 | mixed mode | 0.5039 | 0.2982 | 0.5856 |
| 60d–30–2 | mixed mode | 0.6024 | 0.3565 | 0.7000 |
| 60d–30–3 | mixed mode | 0.6268 | 0.3710 | 0.7284 |
| 60d–54–1 | mode II | — | 0.2944 | 0.2944 |
| 60d–54–2 | mode II | — | 0.3099 | 0.3099 |
| 60d–54–3 | mode II | — | 0.3544 | 0.3544 |

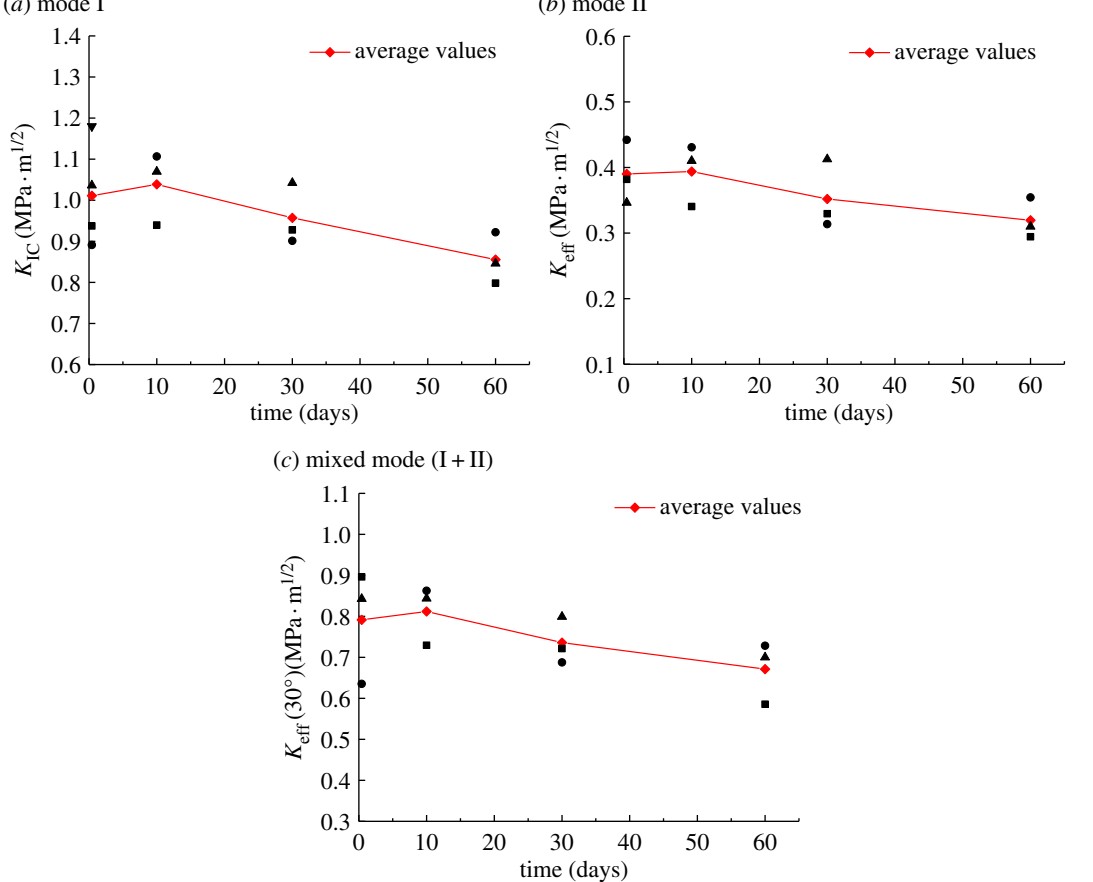

**Figure 9.** Fracture toughness of granite versus thermal treatment duration.

gradual decrease in fracture toughness, reaching $0.3196\,\mathrm{MPa\cdot m^{1/2}}$ for samples maintained at a high temperature for 60 days. This value is 18.09% lower than that for samples heated for only 10 h.

As shown in figure 9c, the mixed-mode (I + II) fracture toughness increased slightly with an increase in the thermal treatment durations from 10 h to 10 days. For heating durations exceeding that, the fracture toughness gradually decreased. The mixed-mode (I + II) fracture toughness of granite decreased from $0.7914\,\mathrm{MPa\cdot m^{1/2}}$ for 10 h of heating to $0.6713\,\mathrm{MPa\cdot m^{1/2}}$ for 60 days, a 15.17% reduction.

It can be seen from figure 9a–c that the mode I, mode II and mixed-mode (I + II) fracture toughness of granite gradually reduced with heating durations. However, the reduction ratio did not exceed 20%, and the reduction was not pronounced.

## 4.2. Test force–displacement curves

The computer recorded force–displacement curves during the fracture toughness tests. Analysis of such curves helps us to understand the fracture characteristics of rock [23]. Some of the test force–displacement curves are shown in figure 10. At the initial stage of loading, the granite undergoes compaction, and the internal pore cracks are gradually closed, giving the curve a relatively shallow slope. However, the compaction stage is not clearly obvious, indicating that the granite is very dense. The granite then enters the elastic stage, deforming elastically. When the elastic limit is reached, it enters the stage of plastic deformation. However, this stage was hardly ever observed: when the granite had been elastically deformed to a certain limit, damage occurred in the form of brittle fracture.

As the duration of thermal treatment increased, the peak load at the time of granite fracture decreased. Therefore, an increase in thermal treatment holding time leads to mechanical weakening of the granite. The slope of the curve gradually decreases, though the decrease is not major. In general, the shape of the force–displacement curve of granite is basically the same after thermal treatment for different periods. In all cases, there is a non-obvious compaction stage and almost no plastic

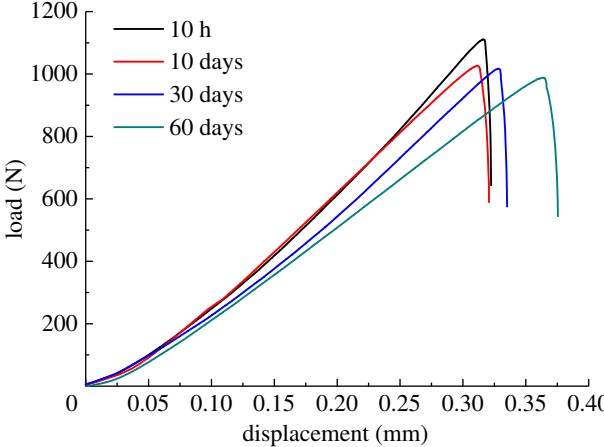

**Figure 10.** Test force–displacement curves.

deformation stage, and the elastic response is extremely strong. The increase in high-temperature holding time did not change the characteristics of the brittle fracturing.

# 5. Discussion

## 5.1. Change in fracture toughness with the duration of heating

The average values for mode I, mode II and mixed-mode (I + II) fracture toughness as a function of temperature duration are plotted in the same coordinate system in figure 11. Table 2 shows a quantitative comparison of changes in the mode I, mode II and mixed-mode (I + II) fracture toughness of granite with the thermal treatment duration. '+' means increase, and '−' means decrease.

It can be seen from figure 11 that the evolution of the fracture toughness of granite with the duration of thermal treatment has two stages. The first stage is where the duration is from 10 h to 10 days and the second is for durations from 10 days to 60 days. With an increase in the duration for which the granite was held at a high temperature, its mode I and mixed-mode (I + II) fracture toughness first increased and then decreased, whereas its pure mode II fracture toughness gradually decreased. The mode I and mixed-mode (I + II) fracture toughness curves of granite show a maximum at 10 days of heat retention, at values of $1.0386\,\mathrm{MPa}\cdot\mathrm{m}^{1/2}$ and $0.8118\,\mathrm{MPa}\cdot\mathrm{m}^{1/2}$, respectively. It can be seen from table 2 that these values equate to increases of +2.72% and +2.58%, respectively, compared with 10 h thermal treatment. The pure mode II fracture toughnesses of granite at 10 h and 10 day thermal treatment are approximately equal. Therefore, there is almost negligible change in the mechanical properties of granite when exposed to high temperature for 10 days or less.

As heating durations exceed 10 days, the mode I, mode II and mixed-mode (I + II) fracture toughnesses of granite gradually decrease, reaching a minimum value when the temperature is maintained for 60 days. Therefore, after thermal treatment duration exceeds 10 days, the mechanical properties of the granite are affected to some extent by increased holding time. The fracture toughness increases with the holding time. After more than 10 days, the curve is a straight downward line, indicating that a continued decrease may occur. Therefore, the ability of the rock to resist fracture failure may be further reduced with a longer holding time (more than 60 days).

It can be seen from figure 11 and table 2 that the mode I fracture toughness value of granite is greater than the pure mode II fracture toughness value, and the mixed-mode (I + II) fracture toughness value is between the two. On the whole, the change trends in the mode I, mode II and mixed-mode (I + II) fracture toughness of granite with a change in the duration of thermal treatment are consistent. That is to say, the ability of granite to resist tensile, shear and mixed (tensile + shear) fracture decreases consistently with an increased duration of heating. Feng *et al.* [24] studied the fracture toughness of sandstone and found that the change trends in mode I, mode II and mixed mode (I + II) with temperatures are similar. They also found that $K_{\mathrm{IIC}} < k_{\mathrm{eff}}\,(\alpha = 30^\circ) < k_{\mathrm{eff}}\,(\alpha = 15^\circ) < K_{\mathrm{IC}}$.

The mode I and mode II fracture toughness of granite increased slightly when kept at a high temperature for 10 days, but the increase was almost negligible. On the whole, the mode I, mode II and mixed-mode (I + II) fracture toughness of granite decreases with increasing thermal treatment

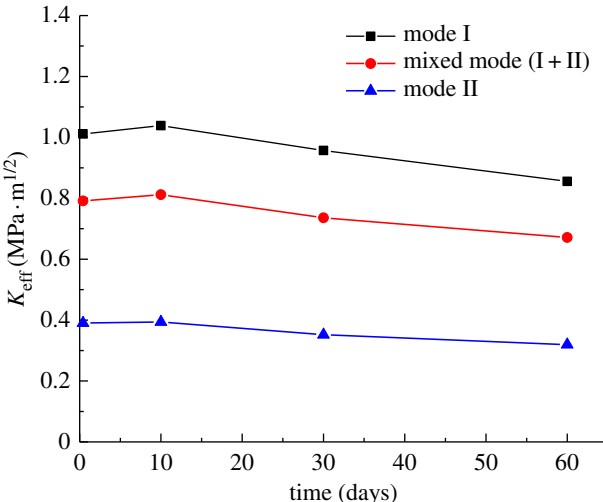

**Figure 11.** Fracture toughness of granite versus thermal treatment duration.

**Table 2.** Changes in fracture toughness over different temperature exposure durations.

| time (days) | mode I (%) | mixed mode (I + II) (%) | mode II (%) |
|---|---|---|---|
| 0.42 − 10 | +2.72 | +2.58 | +0.95 |
| 10 − 60 | −17.64 | −17.31 | −18.84 |

time. During the time period from 10 h to 60 days, the minimum fracture toughness of the granite was less than the maximum, and the reduction ratio did not exceed 20%. Therefore, the change in the fracture toughness of granite is not obvious with a change in the high-temperature holding time.

## 5.2. Effects of temperature on fracture toughness

Changes in the mode I, mode II and mixed-mode (I + II) fracture toughness of granite due to high temperature can be mainly attributed to change in its internal microstructure. When the rock is heated from room temperature to a high temperature, a series of changes occur inside it. First, as the granite heats up, the free water inside it evaporates and its porosity increases. As it lacks the lubrication provided by moisture, the frictional resistance of the granite matrix to applied force increases. As the temperature increases, the adsorbed water inside the rock gradually evaporates. At the same time, the mineral particles thermally expand, which makes the rock more compact. The pore space decreases, and the porosity decreases. When the particles have expanded to a certain extent, they begin to restrain each other. Furthermore, because different mineral particles have different coefficients of thermal expansion, their thermal deformation upon being subjected to temperature is not coordinated, which may cause thermal cracking in the mineral particles. As the temperature continues to rise, the numbers of thermal cracks inside the mineral particles and cracks between the mineral particles increase. At high temperature, the bound water will gradually be removed from the mineral particles. The increasingly pronounced dehydration will cause direct damage to the interior of the rock. A continued increase in temperature will cause mineral-phase transitions, such as the conversion of α-quartz to β-quartz at 573°C. Phase transitions are often the main cause of a sudden increase in internal rock damage via cracking.

Therefore, temperature affects rocks in three main ways. The first is the effect of the temperature rise. When the temperature is raised, a temperature difference is generated, resulting in thermal stress. Thermal stress causes changes in the internal structure of the rock. The second is the effect of the temperature value itself. At different temperatures, the thermal intensity inside the rock is different. Generally, the higher the temperature, the more intense the molecular thermal motion and hence the more frequent the microscopic activity inside the rock. The third is the effect of the time over which the temperature continues to act, that is, the high-temperature holding time. In addition, the essence of

the cooling process is similar to that of the temperature rise, mainly driven by the temperature difference. There must be sufficient time for the occurrence and completion of a series of effects of temperature on the rock, as all of the above microscopic changes in the rock must have sufficient time to ensure adequate completion. For example, the processes of free water evaporation, combined water dehydration, mineral particle expansion, thermal cracking, mineral-phase transformation and the like are all dependent on the duration of temperature action. During the heating process, the rock gradually transfers heat from the outside to the inside. The thermal stress that arises causes the expansion of the mineral particles, and the expansion force is gradually increased from the outside to the inside. The gradual weakening and release of binding force take a certain amount of time to complete. The granite in this paper has a small decrease in fracture toughness after 60 days of heat preservation. It can be seen that long-term heat preservation has caused a certain increase in rock damage. This leads to a reduction in the mode I, mode II and mixed-mode (I + II) fracture toughness of the granite. Moreover, this also shows that the high-temperature rock damage process has a relatively long action time rather than rapid or instantaneous completion.

This paper uses the fracture toughness index and so reflects the influence of temperature exposure time on the ability of rock to resist fracture failure. However, there may be similar laws for other physical and mechanical properties because the physical and mechanical parameters of rock are usually connected to some degree. For example, Chen *et al*. [25] used statistical analysis to study the physical and mechanical parameters of sandstone and shale, and established a relationship between fracture toughness and acoustic wave velocity and dynamic elastic modulus. Brown & Reddish [26] studied the relationship between fracture toughness and density, and Nasseri *et al*. [6] showed that normalized compressional P-wave velocities matched the decreasing trend of normalized fracture toughness ($K_{IC}$) remarkably well. Based on an analysis of experimental data, Jin *et al*. [27] established a relationship between mode II fracture toughness and tensile strength. Bearman [28] studied the method of using a point load test to estimate the mode I fracture toughness quickly. Thus, numerous studies have confirmed that there is a certain correlation between the different mechanical parameters of rock. The duration of high-temperature exposure mainly affects the fracture toughness of rock through changes in its internal mesostructure. This will not only result in changes in the resistance of the granite to fracturing, but may also cause changes in other physical and mechanical parameters. Sirdesai *et al*. [29] subjected red sandstone to thermal treatment with different holding times (5, 10, 15, 20, 25 and 30 days) and then carried out Brazilian splitting experiments. The results show that the highest strength was achieved after 10 days of treatment for almost all temperatures. As the treatment temperature was increased, the peak strength for all treatment durations increased. The results of the current paper show that, under the conditions and sample size obtainable in a laboratory, continuing to subject granite to thermal treatment for more than 10 h has little effect on its fracture resistance.

In experimental studies of the physical and mechanical effects of temperature on rock, it is general practice to maintain the temperature for 2 h or, rarely, more than 8 h. Thermal cracks are generated in rocks under the action of temperature. The thermal cracking of granite is mainly determined by temperature, or it can be said to be caused by the action of thermal stress. The degree to which such cracking occurs mainly depends on the temperature difference. Other factors are secondary and can be ignored [30]. The experimental results of the current paper show that the fracture toughness value changes little after 10 days of heat preservation. Even after the duration of the insulation reached 60 days, the reduction in the fracture toughness value was small. The effect of high-temperature holding time on the ability of rocks to resist fracture damage is therefore judged to be small.

## 6. Conclusion

Granite samples were thermally treated for different periods (10 h, 10 days, 30 days and 60 days), and the SCB method was used to test their mode I, mode II and mixed-mode (I + II) fracture toughness. The effect of thermal treatment duration on the fracture toughness of granite was investigated. The main conclusions obtained are as follows.

(1) The change trend in fracture toughness with the thermal treatment duration is divided into two stages. The first stage is from 10 h to 10 days. Within this range, the mode I and mixed-mode (I + II) fracture toughness of granite showed extremely small increases of 2.72% and 2.58%, respectively. The mode II fracture toughness value did not change. Therefore, heating the granite for 10 days had hardly any effect on its ability to resist fracture. In the second stage, from 10 days to 60 days of thermal treatment, the fracture toughness of the granite gradually reduced. Overall,

the mode I, mode II and mixed-mode (I + II) fracture toughness of granite was reduced by 15.39%, 18.07% and 15.18%, respectively, after 60 days of heating compared with 10 h. Therefore, keeping granite at a high temperature for up to 60 days had little effect on its fracture toughness. Moreover, the brittle fracture characteristics of granite did not change with an increase in high-temperature holding time.

(2) The change trends in the mode I, mode II and mixed-mode (I + II) fracture toughness of granite with the thermal treatment duration are basically consistent. That is, granite shows consistent abilities to resist tensile, shear and mixed (tensile + shear) fracture with change in heating duration. Moreover, under the same heating time, the fracture toughness values rank as pure mode I > mixed mode (I + II) > pure mode II.

(3) The change in the internal microstructure of granite when subjected to high temperature is a long-term process. These microscopic changes are still ongoing and show no signs of stopping when the holding time reaches 60 days. After more than 10 days of high-temperature exposure, the curve of the fracture toughness forms a straight downward-sloping line.

(4) Change in the fracture toughness of granite with change in the heat treatment duration is not obvious overall.

Data accessibility. The datasets supporting this article have been uploaded to the Dryad Digital Repository: https://doi.org/10.5061/dryad.v28t5 [31,32].
Authors' contributions. G.F. performed the experiments, data analyses and manuscript writing. Y.K. designed the experiments and helped to analyse the results. X.-c.W. helped to analyse the results and revised this manuscript.
Competing interests. We declare we have no competing interests.
Acknowledgement. This study was supported by the National Key Basic Research Development Program of China (973 Program) (grant no. 2014CB239200) and the National Natural Science Foundation of China (grant no. 51574173).

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
