## [Reviewer comments · Royal Society Open Science]

Review History

RSOS-190144.R0 (Original submission)

Review form: Reviewer 1 (Haijia Wen)

Is the manuscript scientifically sound in its present form?

Yes

Are the interpretations and conclusions justified by the results?

Yes

Is the language acceptable?

Yes

Is it clear how to access all supporting data?

Yes

Do you have any ethical concerns with this paper?

No

Have you any concerns about statistical analyses in this paper?

No

Recommendation?

Accept with minor revision (please list in comments)

Comments to the Author(s)

Specific comments are as follows:

1. Page 6 line 36, It should be "and" rather than "or".
2. Why are these specimens heated first, and then processed by cutting it artificially?
3. Fracture behavior of granite are widely discussed by many researchers, I suggested the authors make some comparison your results with published articles.
[1] Junfeng Liu, Haiqing Yang, Yang Xiao, Xiaoping Zhou. Macro-mesoscopic Fracture and Strength Character of Pre-cracked Granite Under Stress Relaxation Condition[J]. Rock Mechanics and Rock Engineering, 2018,51(5):1401-1412.
4. Kindly keep a uniform style of referring.
5. Kindly provide high-resolution images for all the attached figures.

Review form: Reviewer 2

Is the manuscript scientifically sound in its present form?

Yes

Are the interpretations and conclusions justified by the results?

Yes

Is the language acceptable?

Yes

Is it clear how to access all supporting data?

Not Applicable

Do you have any ethical concerns with this paper?

No

Have you any concerns about statistical analyses in this paper?

No

Recommendation?

Accept with minor revision (please list in comments)

Comments to the Author(s)

1. The authors shows three mode in figure 2. However, in the manuscript, only mode 1, mode 2 and mode 1+2 appear. Please explain it, or show a sketch of mixed mode. The mode 3 should be revised, it is not readerable.
2. P4 line 15, Most laboratory research currently uses a 2-hour temperature exposure duration, with a few studies using 8 hours. Please add some references here.
3. P4 line 54, "S/D=0.61 and a/R=0.5. With a/R and S/R fixed". Why do the authors just use one of them, S/D or S/R?
4. In this study, $\alpha = 30^\circ$ is selected to explore mixed-mode (I+II) fracture. Why do the authors

choose 30°?

5. Table1, please add the average values and the standard deviations of each group.
6. P8 line 4, "toughness of granite gradually reduced with temperature". I think it changes with heating time, not temperature. And the results do not decrease gradually.
7. P10 line 11, "the free water inside it evaporates". What's the water content of the samples?
8. In figure 9c, Mixed-mode (I+II) fracture toughness ($\alpha = 3^\circ$). What's $\alpha = 3^\circ$ mean?
9. Figure 10, please show us which sample do the authors choose in each group?

Decision letter (RSOS-190144.R0)

01-Apr-2019

Dear Dr Feng

On behalf of the Editors, I am pleased to inform you that your Manuscript RSOS-190144 entitled "The Fracture Failure of Granite after Varied Durations of Thermal Treatment: An Experimental Study" has been accepted for publication in Royal Society Open Science subject to minor revision in accordance with the referee suggestions. Please find the referees' comments at the end of this email.

The reviewers and handling editors have recommended publication, but also suggest some minor revisions to your manuscript. Therefore, I invite you to respond to the comments and revise your manuscript.

- Ethics statement

- Data accessibility

<http://datadryad.org/submit?journalID=RSOS&manu=RSOS-190144>

- Competing interests

- Authors' contributions

- Acknowledgements

- Funding statement

Because the schedule for publication is very tight, it is a condition of publication that you submit the revised version of your manuscript before 10-Apr-2019. Please note that the revision deadline will expire at 00.00am on this date. If you do not think you will be able to meet this date please let me know immediately.

on behalf of Professor R. Kerry Rowe (Subject Editor)
openscience@royalsociety.org

Reviewer comments to Author:

Reviewer: 1

Comments to the Author(s)

Specific comments are as follows:

1. Page 6 line 36, It should be "and" rather than "or".
2. Why are these specimens heated first, and then processed by cutting it artificially?
3. Fracture behavior of granite are widely discussed by many researchers, I suggested the authors make some comparison your results with published articles.
[1] Junfeng Liu, Haiqing Yang, Yang Xiao, Xiaoping Zhou. Macro-mesoscopic Fracture and Strength Character of Pre-cracked Granite Under Stress Relaxation Condition[J]. Rock Mechanics and Rock Engineering, 2018,51(5):1401-1412.
4. Kindly keep a uniform style of referring.
5. Kindly provide high-resolution images for all the attached figures.

Reviewer: 2

Comments to the Author(s)

1. The authors shows three mode in figure 2. However, in the manuscript, only mode 1, mode 2 and mode 1+2 appear. Please explain it, or show a sketch of mixed mode. The mode 3 should be revised, it is not readerable.
2. P4 line 15, Most laboratory research currently uses a 2-hour temperature exposure duration, with a few studies using 8 hours. Please add some references here.
3. P4 line 54, "S/D=0.61 and a/R=0.5. With a/R and S/R fixed". Why do the authors just use one of them, S/D or S/R?
4. In this study, $\alpha = 30^\circ$ is selected to explore mixed-mode (I+II) fracture. Why do the authors choose 30° ?
5. Table1, please add the average values and the standard deviations of each group.
6. P8 line 4, "toughness of granite gradually reduced with temperature". I think it changes with heating time, not temperature. And the results do not decrease gradually.
7. P10 line 11, "the free water inside it evaporates". What's the water content of the samples?
8. In figure 9c, Mixed-mode (I+II) fracture toughness ($\alpha = 3^\circ$). What's $\alpha = 3^\circ$ mean?
9. Figure 10, please show us which sample do the authors choose in each group?

Author's Response to Decision Letter for (RSOS-190144.R0)

See Appendix A.

Decision letter (RSOS-190144.R1)

23-May-2019

Dear Dr Feng,

I am pleased to inform you that your manuscript entitled "The Fracture Failure of Granite after Varied Durations of Thermal Treatment: An Experimental Study" is now accepted for publication in Royal Society Open Science.

on behalf of Prof R. Kerry Rowe (Subject Editor)
openscience@royalsociety.org

Follow Royal Society Publishing on Twitter: [@RSocPublishing](https://twitter.com/RSocPublishing)
Follow Royal Society Publishing on Facebook:
<https://www.facebook.com/RoyalSocietyPublishing.FanPage/>
Read Royal Society Publishing's blog: <https://blogs.royalsociety.org/publishing/>

Appendix A

Dear Editor and Reviewers:

Thank you very much for reviewing and commenting on our paper, “The Fracture Failure of Granite after Varied Durations of Thermal Treatment: An Experimental Study” (Manuscript ID: RSOS-190144). Your comments were valuable and helped us revise and improve this paper; they will also help guide our future research. We have revised our manuscript and made corrections to address your concerns. The revised portions of our manuscript are highlighted in red. The main corrections to the paper and our responses to reviewer comments follow:

Reviewer #1:

Comment 1: Page 6 line 36, It should be "and" rather than "or".

Response: Thank you for your comment. We have corrected this error.

Comment 2: Why are these specimens heated first, and then processed by cutting it artificially?

Response: Thank you for your comment. The process of heating and cooling produces a temperature difference, causing thermal stress. The thermal stress is concentrated at the tip of the artificial notch, causing a fracture toughness test deviation. As such, the heat treatment was done first, followed by artificial cutting.

Comment 3: Fracture behavior of granite are widely discussed by many researchers, I suggested the authors make some comparison your results with published articles.

[1] Junfeng Liu, Haiqing Yang, Yang Xiao, Xiaoping Zhou. Macro-mesoscopic Fracture and Strength Character of Pre-cracked Granite Under Stress Relaxation Condition[J]. Rock Mechanics and Rock Engineering, 2018,51(5):1401-1412.

Response: Thank you for the articles that you recommended. We have read these papers carefully, and compared them to our research. As a result, some supplemental materials have been added to the revised version; they are appropriately cited.

Comment 4: Kindly keep a uniform style of referring.

Response: Thank you for your advice. We have corrected the references in the revised manuscript.

Reviewer #2:

Comment 1: The authors shows three mode in figure 2. However, in the manuscript, only mode 1, mode 2 and mode 1+2 appear. Please explain it, or show a sketch of mixed mode. The mode 3 should be revised, it is not readable.

Response: Thank you for your comment. In this manuscript, only mode-I, mode-II and mixed (I+II) fracture toughness have been studied. The mode-III fracture toughness is one of the basic fracture modes. It has only been shown in this manuscript and has not been studied.

Comment 2: P4 line 15, Most laboratory research currently uses a 2-hour temperature exposure duration, with a few studies using 8 hours. Please add some references here.

Response: Thank you for your advice. We have carefully reviewed this aspect of research. And in the revised draft correctly summed up the research status, while some supplemental materials have been added to the revised version; they are appropriately cited.

Comment 3: P4 line 54, “S/D=0.61 and a/R=0.5. With a/R and S/R fixed”. Why do the authors just use one of them, S/D or S/R?

Response: Thank you for pointing out mistake. The correct statement in this manuscript should be S/D. We read the manuscript carefully and corrected the mistake.

Comment 4: In this study, $\alpha=30^\circ$ is selected to explore mixed-mode (I+II) fracture. Why do the authors choose 30° ?

Response: Thank you for your comment. This study sets $S/D=0.61$ and $a/R=0.5$. According to the research (Lim et al, 1993), when the angle of the pre-notch is $\alpha=54^\circ$, it will be mode II. Other angles will lead to mixed-mode (I+II) fracture. We chooses $\alpha=30^\circ$ is reasonable and is conducive to sample processing.

Comment 5: Table 1, please add the average values and the standard deviations of

each group.

Response: Thank you for your comment. Table 1 provides the basic data of the experiment. The average of the experimental data is given in Figure 9. The data is analyzed by the average value. Therefore, it is not necessary to list the standard deviations of each group in Table 1.

Comment 6: P8 line 4, “toughness of granite gradually reduced with temperature”. I think it changes with heating time, not temperature. And the results do not decrease gradually.

Response: Thank you for pointing out mistake. The correct statement in this manuscript should be heating durations. We read the manuscript carefully and corrected the mistake.

Comment 7: P10 line 11, “the free water inside it evaporates”. What’s the water content of the samples?

Response: Thank you for your comment. The natural moisture content of granite is 0.6%. The granite used in this manuscript is taken from the natural environment. As the temperature rises, the water in the rock gradually evaporates. As it lacks the lubrication provided by moisture, the frictional resistance of the granite matrix to applied force increases.

Comment 8: In figure 9c, Mixed-mode (I+II) fracture toughness ($\alpha=3^\circ$). What’s $\alpha=3^\circ$ mean?

Response: Thank you for pointing out mistake. The correct angle should be 30° . We read the manuscript carefully and corrected the mistake.

Comment 9: Figure 10, please show us which sample do the authors choose in each group?

Response: Thank you for your comment. From 10h to 60 days of heating durations, the curves are selected from 300-10h-0-3, 300-10d-0-1, 300-30d-0-2, 300-60d-0-3.